

# Integrin-associated transcriptional characteristics of circulating tumor cells in breast cancer patients

Evgeniya Grigoryeva[1,2], Liubov Tashireva[1,3], Vladimir Alifanov[3], Olga Savelieva[3], Marina Zavyalova[3], Maxim Menyailo[4], Anna Khozyainova[4], Evgeny V. Denisov[4], Olga Bragina[5], Nataliya Popova[6], Nadezhda V. Cherdyntseva[2] and Vladimir Perelmuter[3]

[1] The Laboratory of Molecular Therapy of Cancer, Cancer Research Institute, Tomsk National Research Medical Center, Russian Academy of Sciences, Tomsk, Russia
[2] The Laboratory of Molecular Oncology and Immunology, Cancer Research Institute, Tomsk National Research Medical Center, Russian Academy of Sciences, Tomsk, Russia
[3] The Department of General and Molecular Pathology, Cancer Research Institute, Tomsk National Research Medical Center, Russian Academy of Sciences, Tomsk, Russia
[4] The Laboratory of Cancer Progression Biology, Cancer Research Institute, Tomsk National Research Medical Center, Russian Academy of Sciences, Tomsk, Russia
[5] The Department of Nuclear Therapy and Diagnostics, Cancer Research Institute, Tomsk National Research Medical Center, Russian Academy of Sciences, Tomsk, Russia
[6] The Department of Chemotherapy, Cancer Research Institute, Tomsk National Research Medical Center, Russian Academy of Sciences, Tomsk, Russia

Corresponding author
Evgeniya Grigoryeva,
grigoryeva.es@gmail.com

## ABSTRACT

**Background:** Integrins enable cell communication with the basal membrane and extracellular matrix, activating signaling pathways and facilitating intracellular changes. Integrins in circulating tumor cells (CTCs) play a significant role in apoptosis evasion and anchor-independent survival. However, the link between CTCs expressing different integrin subunits, their transcriptional profile and, therefore, their functional activity with respect to metastatic potential remains unclear.

**Methods:** Single-cell RNA sequencing of CD45-negative cell fraction of breast cancer patients was performed. All CTCs were divided into nine groups according to their integrin profile.

**Results:** CTCs without the gene expression of integrins or with the expression of non-complementary α and β subunits that cannot form heterodimers prevailed. Only about 15% of CTCs expressed integrin subunits which can form heterodimers. The transcriptional profile of CTCs appeared to be associated with the spectrum of expressed integrins. The lowest potential activity was observed in CTCs without integrin expression, while the highest frequency of expression of tumor progression-related genes, namely genes of stemness, epithelial-mesenchymal transition (EMT), invasion, proinflammatory chemokines and cytokines as well as laminin subunits, were observed in CTCs co-expressing *ITGA6* and *ITGB4*. Validation on the protein level revealed that the median of integrin β4+ CTCs was higher in patients with more aggressive molecular subtypes as well as in metastatic breast cancer patients. One can expect that CTCs with *ITGA6* and *ITGB4* expression will have pronounced metastatic potencies manifesting in expression of EMT and

stemness-related genes, as well as potential ability to produce chemokine/proinflammatory cytokines and laminins.

## INTRODUCTION

Integrins are adhesion receptors, which are polyfunctional molecules that ensure the connection between the cells and extracellular matrix (ECM). Functionally active integrins are represented by heterodimers made up of α and β subunits arranged in numerous dimeric pairings. These pairings are known to involve 18 α and 8 β subunits resulting in the formation of 24 αβ heterodimeric receptors. The activation of integrins increases the affinity of these molecules to extracellular ligands (*Mezu-Ndubuisi & Maheshwari, 2021*). The binding of integrins to ligands on the basal membrane and extracellular matrix is critical for most intracellular events, both under normal and pathological conditions (*Su et al., 2020*).

In tumors, interactions *via* integrins between tumor cells and microenvironment trigger multiple signaling pathways and contribute to the malignant phenotype of carcinoma cells. The role of integrins in key processes of cancer progression, such as invasion, intravasation, acquisition of stem cell-like properties, extravasation at distant sites, and resistance to therapeutic agents is well-studied (*Su et al., 2020*; *Hamidi & Ivaska, 2018*; *Cooper & Giancotti, 2019*). Recent studies demonstrate the important role that integrins play on tumor-derived exosomes in the organotropic metastasis of breast cancer (*Hoshino et al., 2015*). However, data on the expression of integrins in CTCs are extremely limited.

Our previous study demonstrated a lack of correlation between integrin expression on primary tumor cells and CTCs (*Grigoryeva et al., 2023*). Indeed, in contrast to the cancer cells at the primary site, CTCs lose their ability to interact with ligands of the basal membrane and intercellular matrix. It is known that integrins are expressed on CTCs and play an essential role in the survival of tumor cells in circulation providing an anchor-independent survival (*Hamidi & Ivaska, 2018*). Among the integrins involved in carcinoma progression, integrin α6β4 plays a prominent role. While the significance of integrin α6β4 expression on primary tumor cells in breast cancer has been established in many studies, the study of this integrin in CTCs is extremely limited. *Sharifi et al. (2021)* showed the association of α6β4 integrin expression on CTCs with metastasis predominantly to the bones. The authors claim that this is the first publication devoted to the study of α6β4 in CTCs in breast cancer (*Sharifi et al., 2021*).

The objective of our study was to establish the integrin profile of CTCs and determine its correlation with their transcriptional characteristics, thereby elucidating the potential association between functional potencies of CTCs and risk of distant metastases. Portions of this text were previously published as part of *Menyailo et al. (2023)*.

## MATERIALS AND METHODS

### Patients

The study involved 81 female individuals with non-metastatic invasive breast carcinoma (BC) of no special type (T1-4N0-3M0, covering all molecular subtypes) who received treatment at the Cancer Research Institute, Tomsk National Research Medical Center between 2019 and 2021. Among these patients, 24.7% (20 out of 81) demonstrated the presence of CD45-EpCAM+/KRT7/8+ CTCs (eight cells per 1 ml of blood) and were subsequently included in the single-cell RNA sequencing (scRNA-seq) analysis. Table S1 provides details regarding the clinical and pathological parameters of these patients. The study received approval from the local ethics committee of the Tomsk NRMC Institutional Review Board on June 17, 2016, with approval No. 8, and all patients granted informed consent before participating in the analysis. The patients received treatment in accordance with the ESMO Clinical Practice Guidelines (https://www.esmo.org/guidelines/breast-cancer/early-breast-cancer).

### Blood collection and preparation

Sample preparation were performed as previously described in (*Menyailo et al., 2023*). Venous blood samples (15 ml) were collected either before the administration of neoadjuvant chemotherapy or one to 2 days prior to surgical intervention (for patients not undergoing neoadjuvant chemotherapy) using EDTA tubes. Aliquots for flow cytometry analysis (3 mL) were incubated at 37 °C for 1.5 h. Subsequently, white blood cells were carefully aspirated from the thin, intermediate layer between the plasma and red blood cells after their sedimentation. The acquired cell concentrate was washed with 2 mL of Cell Wash buffer (BD Biosciences, San Jose, CA, USA) through centrifugation at 800 × g for 15 min.

Blood samples preparation for scRNA-seq procedure was performed as follows: CD45−negative cell fraction was enriched using negative selection by depletion of CD45+, glycophorin A+, and CD66b+ cells (RosetteSep Human CD45 Depletion Cocktail, STEMCELL Technologies Inc., Canada). Blood samples were incubated with RosetteSep Cocktail (50 µl per 1 ml blood) at RT for 20 min, diluted at 1:1 ratio with 1X phosphate-buffered saline (PBS), and transferred equally into separate Falcon conical 15 ml tubes, containing lymphosep-1077 (Biowest, Riverside, MO, USA) density gradient media. After density gradient centrifugation (1,200 × g for 20 min, break off), the peripheral blood mononuclear cell layer was isolated and washed with RPMI-1640 medium containing 5% fetal bovine serum (FBS) (300 × g for 10 min). Platelets were removed by two additional centrifugations at 110 × g for 10 min at RT with no brake. Obtained cell pellets were resuspended in 100 µl of RPMI-1640 medium containing 20% FBS and 15% dimethylsulfoxide (DMSO), frozen at −80 °C, and stored not more than 1 month before scRNA-seq procedure.

## Flow cytometry

### CTCs detection

The screening of patients' blood samples, suitable for subsequent single-cell RNA sequencing, was conducted using flow cytometry. CTCs were labeled with the following antibodies and fluorescent dyes: BV650-anti-EpCAM (clone 9C4, mouse IgG2b, Sony Biotechnology, San Jose, CA, USA), Alexa Fluor 647-anti-cytokeratin 7/8 (clone CAM5.2, Mouse IgG2a, BD Pharmingen, San Jose, CA, USA), and APC-Cy7-anti-CD45 (clone HI30, mouse IgG1, Sony Biotechnology, San Jose, CA, USA). For intracellular staining, cells were permeabilized using BD Cytofix/Cytoperm (BD Biosciences, San Jose, CA, USA). Isotype control antibodies at the same concentration were added to the control sample. CTCs were identified as CD45-EpCAM+/KRT7/8+ cells. MCF-7 and U937 cells served as positive and negative controls, respectively. The phenotypic characteristics of CTCs were assessed using the Novocyte 3000 flow cytometer (ACEA Biosciences, San Diego, USA). Samples with more than eight CTCs with CD45-EpCAM+CK7/8- and/or CD45-EpCAM-CK7/8+ and/or double-positive CD45-EpCAM+CK7/8+ phenotypes per 1 ml of peripheral blood were selected for subsequent scRNA-seq.

### CTCs immunophenotyping

CTCs immunophenotyping were performed as described in our previous study (*Grigoryeva et al., 2023*). Surface markers (CD45, EpCAM (CD326), integrins β3, β4, and αVβ5) were stained in the initial step, followed by intracellular staining during the subsequent step. The procedure began with a 10-min incubation with 5 µL of Fc Receptor Blocking Solution (Human TruStain FcX, Sony Biotechnology, San Jose, CA, USA) at the room temperature. Following this, monoclonal antibodies were added and incubated at room temperature for 20 min: APC-Cy7-anti-CD45 (clone HI30, IgG1, Sony Biotechnology, San Jose, CA, USA), BV 650-anti-EpCAM (clone 9C4, IgG2b, Sony Biotechnology, San Jose, CA, USA), BV 421-anti-β3 integrin (clone VI-PL2, BD Biosciences, Franklin Lakes, NJ, USA), Alexa Fluor 488-anti-β4 integrin (clone 422325, R&D Systems, Minneapolis, MN, USA), BV Alexa Fluor 750-anti-αVβ5 integrin (clone P5H9, R&D Systems, Minneapolis, MN, USA), PE-Cy7-anti-N-cadherin (clone 8C11, mouse IgG1, Sony Biotechnology, San Jose, CA, USA), BV 510-anti-CD44 (clone G44-26, IgG2b, BD Biosciences, San Jose, CA, USA), PerCP-Cy5.5-anti-CD24 (clone ML5, IgG2a, Sony Biotechnology, San Jose, CA, USA), BV 786-anti-CD133 (clone 293C3, IgG2b, BD Biosciences, San Jose, CA, USA). After incubation, red blood cells were lysed using 250 µL of OptiLyse C buffer (Beckman Coulter, Villepinte, France) at room temperature for 10 min in the dark, and the obtained cell suspension was washed with 1 mL of Cell Wash buffer (BD Biosciences, San Jose, CA, USA) through centrifugation at 800 × g for 6 min.

For intracellular staining, the cells underwent permeabilization using 250 µL of BD Cytofix/Cytoperm (BD Biosciences, San Jose, CA, USA) at 4 °C for 30 min in the dark and were subsequently washed twice in 1 mL of BD Perm/Wash buffer (BD Biosciences, San Jose, CA, USA) at 800 × g for 6 min. The samples were then diluted in 50 µL of BD Perm/Wash buffer (BD Biosciences, San Jose, CA, USA) and incubated at 4 °C for 10 min

in the dark, with the addition of 5 µL of Fc Receptor Blocking Solution (Human TruStain FcX, Sony Biotechnology, San Jose, CA, USA).

Subsequently, monoclonal antibodies BV 650-anti-EpCAM (clone 9C4, IgG2b, Sony Biotechnology, San Jose, CA, USA) and PE-anti-ALDH1A1 (clone 03, IgG1, Sino Biological, Beijing, China) were added and incubated at 4 °C for 20 min. Following incubation, the samples were washed in 1 mL of Cell Wash buffer (BD Biosciences, San Jose, CA, USA) at 800 × g for 6 min and then diluted in 100 µL of Stain buffer (Sony Biotechnology, San Jose, CA, USA). Both unstained control and antibody quality control assessments were performed. The appropriate isotype antibodies were added into the isotype control sample at equivalent concentrations. Compensation beads (VersaComp Antibody Capture Bead kit, Beckman Coulter, Brea, CA, USA) were utilized for compensation control. Flow cytometry analysis was conducted using the Novocyte 3000 (ACEA Biosciences, San Diego, CA, USA).

The gating strategy was as follow debris and doublets were initially discriminated using forward (FSC) and side scatter (SSC) parameters. Subsequent analysis was limited to CD45-negative and EpCAM-positive cells. Expression of integrins β3, β4, and αVβ5 in CTCs was assessed.

## Single-cell RNA sequencing

Cell samples were thawed at 37 °C for 2 min using a water bath. Then, 1 ml of warm RPMI-1640 medium with 10% FBS was added to cells. After centrifuging (300 × g for 5 min), the supernatant was carefully collected, washed in 50 ul PBS with 0.04% BSA by pipetting with wide-bore pipette tips, and placed on the ice. Cell counting was carried out using 0.4% trypan blue (Thermo Fisher Scientific, Waltham, MA, USA) staining in hemocytometer.

Single-cell cDNA libraries were prepared using the Single Cell 3′ Reagent Kit v3.1 and a 10x Genomics Chromium Controller. The number of cells in each channel of the Single-Cell Chip G varied from 3,300 to 10,000. The concentration of cDNA libraries was measured by the dsDNA High Sensitivity kit on a Qubit 4.0 fluorometer (Thermo Fisher Scientific, Waltham, MA, USA). The quality of cDNA libraries was assessed using High Sensitivity D1000 ScreenTape on a 4150 TapeStation (Agilent, Santa Clara, CA, USA). The ready cDNA libraries were pooled, denatured, and sequenced on NextSeq 500 and NextSeq 2000 (Illumina, San Diego, CA, USA) using pair-end reads (28 cycles for read 1, 91 cycles for read 2, and 8 cycles for i7 index). An online implementation of RNASeqPower was used (https://rodrigo-arcoverde.shinyapps.io/rnaseq_power_calc/). The calculated statistical power of this experimental design is 0.5917 (depth = 135156, cv = 1, effect = 2, $n$ = 20, alpha = 0.05). The utilization of biological and technical replicates was not feasible due to the exceedingly rare detection of CTCs in the blood of patients.

The Molecular Signatures Database (MSigDB) v7.0 was used to identify the molecular processes activated in CTCs (*Liberzon et al., 2015*). The scRNA-seq data generated for this study are available *via* BioProject under the accession number PRJNA776403.

## Statistical analysis

The data was analyzed using the GraphPad Prism 9 (GraphPad Software, San Diego, CA, USA). The two-way ANOVA was used for multiple comparison between independent groups; the sample was also checked for outliers. $p < 0.05$ was considered statistically significant.

## RESULTS

### Detection of CTCs

A total of 81 non-metastatic breast cancer (BC) patients were included in the study. Blood samples of 20 BC patients (24.7%) which contained more than 8 CTCs/ml with CD45-EpCAM+CK7/8- and/or CD45-EpCAM-CK7/8+ and/or double-positive CD45-EpCAM+CK7/8+ phenotypes were used for further enrichment procedure and scRNA-seq.

In total, scRNA-seq identified 42,225 cells and all data was consolidated into one file. The reclustering workflow was initiated with the corresponding parameters (threshold by UMIs and features and mitochondrial UMIs), resulting in 67.5% of removed barcodes. CTCs were identified based on the gene expression of one or more epithelial markers: *KRT5*, *7*, *8*, *14*, *18*, *EPCAM*, *MUC1* and *CDH1* (E-cadherin). Thus, the number of CTCs was represented by 445 cells in all samples and gene expression of integrin subunits was evaluated in each cell. A total of 14 integrin subunits were found to be expressed in 445 CTCs: nine α (*ITGA2*, *ITGA2b*, *ITGA4*, *ITGA5*, *ITGA6*, *ITGAE*, *ITGAL*, *ITGAM*, and *ITGAV*) and five β subunits (*ITGB1*, *ITGB2*, *ITGB3*, *ITGB4*, and *ITGB5*). The subunit co-expression in each CTC was evaluated, and if there were complementary α and β subunits, the potential presence of heterodimeric receptor in the cell was supposed. Depending on the combinations of potential heterodimers and subunits in the CTCs, cells were divided into nine groups represented in Table 1.

The group 1 included 143 CTCs expressing non-complementary integrin α- and β-subunits. 81 CTCs with only integrin α-subunits expression were included in group 2. Groups 3–5 included CTCs with *ITGB1*, *ITGB2*, and *ITGB3* gene expression with complementary α-subunits and no other β-subunits, respectively. CTCs with gene expression of different β-subunits and containing at least one potential heterodimer were represented in group 6. CTCs with *ITGB4* gene expression without complementary α-subunits formed group 7, while cells with *ITGB4* and complementary *ITGA6* expression were included in group 8. These cells are further referred to as CTCs with a potentially full-fledged integrin α6β4 receptor. Group 9 included CTCs without expression of integrin subunits (Table 1).

### Transcriptional characteristics of CTCs with different combinations of integrins subunits gene expression

We evaluated the Top50 differential gene expression in groups with different combinations of integrin subunit expression compared to the no integrins CTCs using Loupe Browser. The only group with differential gene expression was CTCs belonging to group 8 expressing *ITGB4* and complementary *ITGA6*. The following genes were upregulated: *WWTR1*, *CAV1*, *VWF*, *IGFBP7*, *POSTN*, *FN1*, *COL3A1*, *ANXA2*, *MMRN1*, *MGP*, *TIMP1*,

**Table 1 Groups of CTCs depending on combination of integrin subunits gene expression.**

| Genes | Integrin subunits and potential heterodimers | CTC number | Number of significant differences in expression of tumor progression-related genes compared to the CTCs without integrins (group 9) |
|---|---|---|---|
| 1. CTCs expressing non-complementary integrin subunits | | $n = 143$ | 2/36 |
| 2. CTCs expressing only integrin α-subunits | | $n = 81$ | 5/36 |
| 3. CTCs expressing different heterodimers containing β1-subunit without other β-subunits | | $n = 8$ | 2/36 |
| *ITGB1, ITGA2* | α2β1 | $n = 1$ | |
| *ITGB1, ITGA4, ITGAE* | α4β1, αE | $n = 2$ | |
| *ITGB1, ITGA6* | α6β1 | $n = 4$ | |
| *ITGB1, ITGAV* | αVβ1 | $n = 1$ | |
| 4. CTCs expressing different heterodimers containing β2-subunit without other β-subunits | | $n = 2$ | 5/36 |
| *ITGB2, ITGAM* | αMβ2 | $n = 1$ | |
| *ITGB2, ITGAM, ITGAL* | αMβ2 αLβ2 | $n = 1$ | |
| 5. CTCs expressing different heterodimers containing β3-subunit without other β-subunits | | $n = 11$ | 1/36 |
| *ITGB3, ITGA2b* | α2bβ3 | $n = 9$ | |
| *ITGB3, ITGA2b, ITGA5* | α2bβ3, α5 | $n = 1$ | |
| *ITGB3, ITGA2, ITGAE* | α2bβ3, αE | $n = 1$ | |
| 6. CTCs expressing different β-subunits with at least one potential heterodimer | | $n = 38$ | 5/36 |
| *ITGB1, ITGB2, ITGAE, ITGA2b, ITGA4* | α4β1, β2, αE, α2b | $n = 1$ | |
| *ITGB1, ITGB2, ITGAE, ITGAL* | αLβ2, β1 | $n = 1$ | |
| *ITGB1, ITGB2, ITGB3, ITGAE, ITGAL* | αLβ2, β1, β3, αE | $n = 1$ | |
| *ITGB1, ITGB3, ITGB5, ITGA2B* | α2bβ3, β1, β5 | $n = 9$ | |
| *ITGB1, ITGB3, ITGB5, ITGA2B, ITGA6* | α2bβ3, α6β1, β1, β5 | $n = 1$ | |
| *ITGB1, ITGB3, ITGB5, ITGA2B, ITGA2* | α2bβ3, α2β1, β5 | $n = 2$ | |
| *ITGB1, ITGB3, ITGB5, ITGA2B, ITGAE* | α2bβ3, β1, β5, αE | $n = 1$ | |
| *ITGB1, ITGB3, ITGB5, ITGA2B, ITGAM* | α2bβ3, β1, β5, αM | $n = 1$ | |
| *ITGB1, ITGB3, ITGB5, ITGA2B, ITGA2, ITGA6, ITGAE, ITGAM* | α2bβ3, α2β1, α6β1, β5, αE, αM | $n = 1$ | |
| *ITGB1, ITGB3, ITGB5, ITGA2, ITGA5* | α2β1, α5β1, β3, β5 | $n = 1$ | |
| *ITGB1, ITGB3, ITGA2B* | α2bβ3, β1 | $n = 13$ | |
| *ITGB1, ITGB3, ITGA2B, ITGA6* | α2bβ3, α6β1 | $n = 2$ | |
| *ITGB1, ITGB3, ITGA2B, ITGAV, ITGAM* | α2bβ3, β1, αV, αM | $n = 1$ | |
| *ITGB1, ITGB3, ITGA6* | α6β1, β3 | $n = 1$ | |
| *ITGB3, ITGB5, ITGA2B* | α2bβ3 β5 | $n = 2$ | |

| Genes | Integrin subunits and potential heterodimers | CTC number | Number of significant differences in expression of tumor progression-related genes compared to the CTCs without integrins (group 9) |
|---|---|---|---|
| 7. CTCs expressing β4-subunit along with other subunits or heterodimers | | n = 3 | 5/36 |
| *ITGB4* | β4 | n = 1 | |
| *ITGB4*, ITGB2 | β4, β2 | n = 1 | |
| *ITGB4*, ITGB1, TGA4, ITGAE | **α4β1,** β4, αE | n = 1 | |
| 8. CTCs expressing genes of full-fledged integrin α6β4 | | n = 5 | 26/36 |
| *ITGB4*, ITGB1, ITGA6, ITGAE, ITGAV | **α6β4, α6β1, αvβ1,** αE | n = 1 | |
| *ITGB4*, ITGB1, ITGB3, ITGB5, ITGA2, ITGA5, ITGA6, ITGAV | **α6β4, α6β1, αvβ1, α2β1, α5β1, αvβ3, αvβ5** | n = 1 | |
| *ITGB4*, ITGB1, ITGA2, ITGA5, ITGA6, ITGAV | **α6β4, α6β1, αvβ1, α2β1, α5β1** | n = 1 | |
| *ITGB4*, ITGB1, ITGA2, ITGA5, ITGA6, ITGAE, ITGAV | **α6β4, α6β1, αvβ1, α2β1, α5β1** αE | n = 2 | |
| 9. CTCs without integrins gene expression | | n = 154 | |

*LMNA, VIM, KLF6* ($p < 0.05$). To identify the molecular processes activated in these cells we used the Molecular Signatures Database (MSigDB) v7.0 (*Liberzon et al., 2015*) (Fig. 1). It turned out that the largest number of overexpressed genes (6/14) was involved in the EMT process ($p = 2.93\text{e}{-}11$).

The main goal of the study was to assess the hypothesis that the expression of different integrins could be associated with expression of genes which are relevant to the most important functions that determine the aggressive potential of tumor cells: stemness, EMT, chemotaxis, invasion, proliferative activity, resistance to apoptosis, and synthesis of proinflammatory factors. A complete list of the above-mentioned genes is included in Tables S2–S5. We hypothesized that the increased proportion of cells expressing these genes in groups 1–8 compared with CTCs that did not express integrins (group 9) will indicate an association of functional potencies of these cells with expression of certain integrin subunits or heterodimers. The most pronounced differences in the frequencies of tumor progression-related gene expression were observed in the CTCs expressing *ITGB4* with complementary *ITGA6* (group 8). These CTCs essentially differ from cells expressing other combinations of integrin subunit genes (groups 1–7) in three characteristics.

The first is that only in these CTCs, the gene expression of the greatest number of potential heterodimers of integrins (3 to 7) was observed. Thus, five CTCs with the co-expression of *ITGA6* and *ITGB4* (group 8) simultaneously expressed three to seven other potential heterodimers, whereas among the remaining 286 CTCs expressing other integrins, only one cell could express three heterodimers ($p = 0.0000$).

The second was the composition of heterodimers in CTCs belonging to group 8 differs from that in other groups. Subunits that could form potential heterodimers α6β4, αvβ1,

**Figure 1 Gene Set Enrichment Analysis of Top50 genes in CTCs with co-expression of *ITGA6* and *ITGB4* (MSigDB, H1).** The list contains 14 overexpressed genes in CTCs with co-expression of *ITGA6* and *ITGB4*. The largest number of overexpressed genes (6/14) corresponds to epithelial-mesenchymal transition (EMT).

and αvβ5 were found only in CTCs belonging to group 8, whereas α4β1, αMβ2, αLβ2, and α2bβ3 were detected in cells belonging to groups 3–7 in which there was no integrin α6β4 gene expression. The genes of subunits forming α2β1, α6β1, αVβ1, and α5β1 heterodimers were expressed in CTCs independently of the presence of *ITGA6* and *ITGB4* subunits.

The third and the most significant difference is that CTCs co-expressing the *ITGA6* and *ITGB4* genes (group 8) were characterized by a more pronounced potential functional activity than other CTCs (groups 1–7). It should be emphasized that CTCs expressing only the *ITGB4* do not demonstrate similar properties in comparison with no integrins CTCs. Among CTCs with the expression of heterodimeric integrin receptor containing the β4-subunit, a higher frequency of gene expression of ligands for integrin α6β4 receptor (*LAMA4*, *LAMA5*, *LAMB1*, *LAMB2*, *LAMC1*) and stemness genes (*CD44/CD24*, *ALDH1*, *CD133*, *KLF4*, *MYC*) compared with no integrin CTCs was noted (Fig. 2; Tables S2 and S3). The expression of *TGFB1* in these CTCs makes autocrine and paracrine initiation of EMT possible (Table S4). Indeed, traits of EMT were observed in these cells, namely, the combination of epithelial features (positive gene expression of *KRT8*, *10*, *18*, *19* and *MUC1*) with mesenchymal features (negative expression of EPCAM and positive

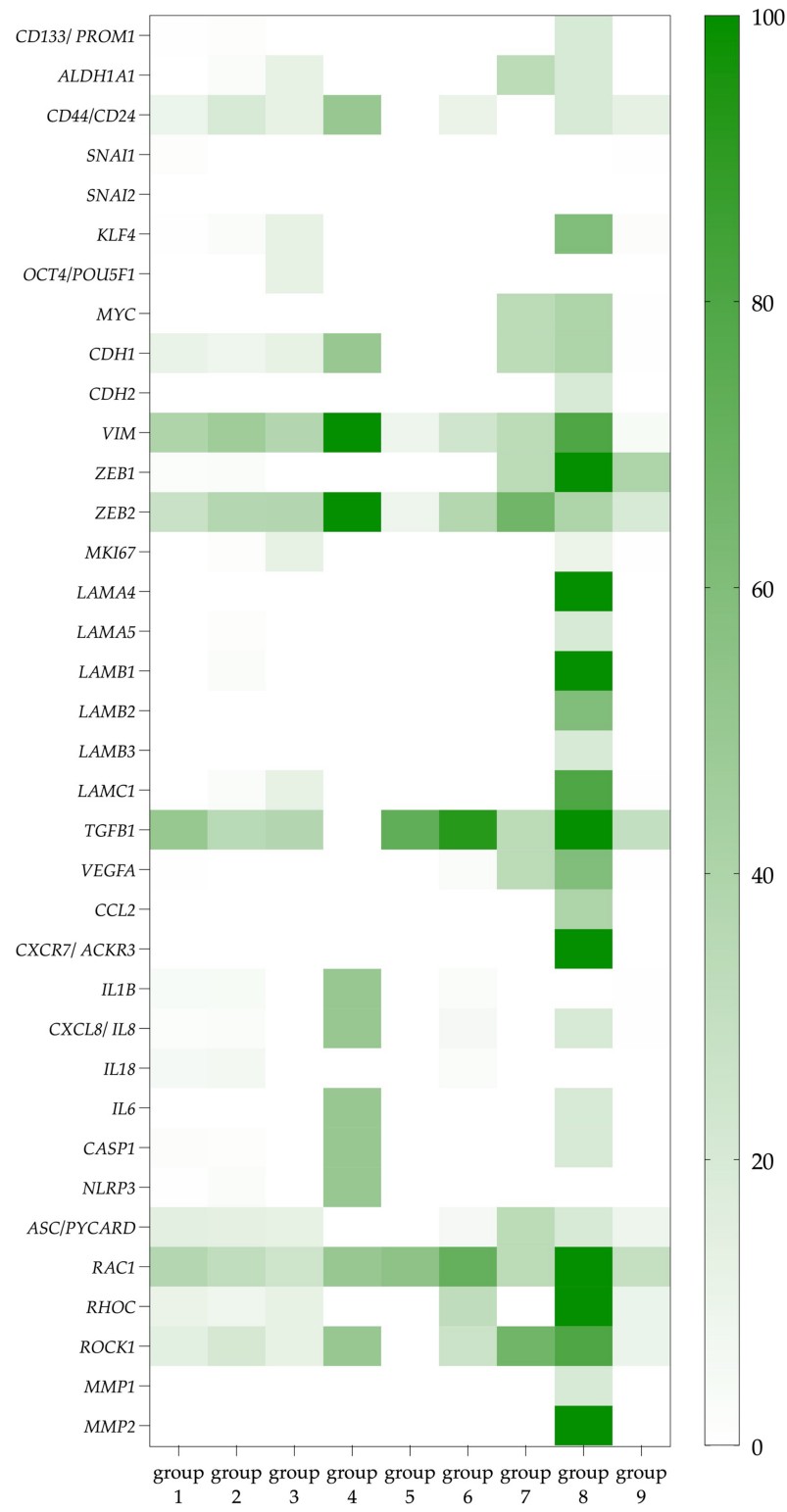

**Figure 2 The proportion of CTCs with positive genes of interest expression in each studied group.** Data represented as a percentage of the total number of CTCs in each group, which are taken as 100 percent. The group 1 included CTCs expressing non-complementary integrin α- and β-subunits. CTCs with only integrin α-subunits expression were included in group 2. Groups 3–5 included CTCs with *ITGB1*, *ITGB2*, and *ITGB3* gene expression with complementary α-subunits and no other β-subunits,

**Figure 2** (continued)
respectively. CTCs with gene expression of different β-subunits and containing at least one potential heterodimer were represented in group 6. CTCs with *ITGB4* gene expression without complementary α-subunits formed group 7, while cells with *ITGB4* and complementary *ITGA6* expression were included in group 8. These cells are further referred to as CTCs with a potentially full-fledged integrin α6β4 receptor. Group 9 included CTCs without expression of integrin subunits.

expression of *CDH2*, *VIM*, *ZEB1*) (Tables S2 and S6). This suggests that CTCs with a potential full-fledged integrin α6β4 receptor have a hybrid EMT phenotype. Taking into account the expression of the *RAC1*, *RHOC*, *ROCK*, *MMP1* and *MMP2*, it appears that this group of CTCs is capable of a predominantly mesenchymal type of invasion (Table S5).

In addition to the characteristics that designate CTCs co-expressing the *ITGA6* and *ITGB4* genes as prospective metastatic seeds, these CTCs exhibited phenotypic traits reminiscent of cells involved in the formation of pre-metastatic niches. Particularly, these CTCs most frequently expressed *VEGFA*, a factor contributing to the formation of pre-metastatic niches. The more frequent expression of factors contributing to inflammation is also important both for the formation of the microenvironment in the primary tumor site and during the formation of pre-metastatic niches. Such factors include the chemokine genes (*IL6*, *MCP1* (CCL2)) which could attract not only macrophages and leukocytes, but also bone marrow progenitor cells. Also, more frequent expression of the chemokine receptor gene *ACKR3* (CXCR7) which binds SDF-1 may provide tumor cell adhesion in the pre-metastatic niche.

Thus, the expression of 26/36 tumor progression-related genes is more frequently observed in cells potentially expressing integrin α6β4 compared to the no integrins CTCs. This result is highlighted by the absence of significant differences between the frequency of gene expression in the other groups with different combinations of integrin subunit expression (only 1–6 out of the 36 genes) (Table 1).

To summarize, CTCs expressing *ITGA6* and *ITGB4* are distinguished by the expression of the greatest number and diversity of other complementary integrin subunits, which could potentially form the functionally active dimers. Only CTCs with the indicated integrin profile had such pronounced functional potencies, namely, the synthesis of laminins, which are the ligands for integrin α6β4, stemness which was manifested by the expression of *CD133*, *ALDH1A1*, *CD44/CD24*, as well as the EMT hybrid phenotype, mesenchymal mechanism of movement/invasion and the synthesis of chemokines and pro-inflammatory cytokines. Therefore, CTCs with the *ITGA6* and *ITGB4* expression have a set of potencies that could lead to tumor cells spread across the body.

## Immunophenotyping of integrin β4+ CTCs in breast cancer patients

Molecular subtypes in BC have been correlated with differences in metastasis rates. Luminal B (HER2+) is known to have the worst prognosis among the luminal subtypes, whereas the triple-negative subtype has the highest risk of metastases among all subtypes. We assessed the protein expression of integrin β4 on CTCs in breast cancer patients with different molecular subtypes. The median of integrin β4+ CTCs in patients with luminal

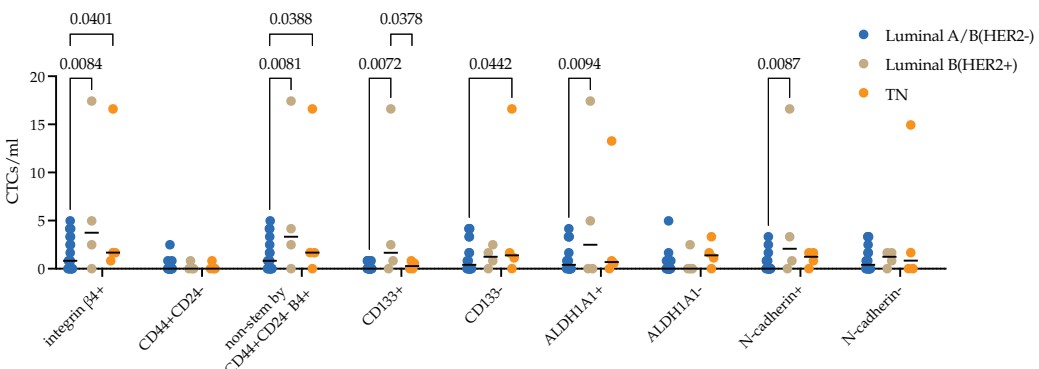

**Figure 3 Stem and EMT properties of integrin β4+ CTCs in breast cancer patients.** The median of integrin β4+ CTCs in patients with luminal A/B (HER2−), luminal B (HER2+) and TNBC was 0.83 (0.00; 3.32), 3.74 (0.62; 14.32) and 1.67 (1.04; 12.87) cell/ml. Number of integrin β4+ CTCs was significantly higher in luminal B (HER2+) and TNBC patients compared to luminal A/B (HER2−) ($p = 0.0084$ and $p = 0.0401$, respectively). The pattern found was provided by non-stem by CD44+CD24-, stem by ALDH1A1+ and CD133+ CTCs and CTCs during EMT (N-cadherin+) in luminal B (HER2+) subtype. Whereas in the triple-negative (TN) type, differences were ensured by non-stem by CD44+CD24- stem CTCs.

A/B (HER2−), luminal B (HER2+) and TNBC was 0.83 (0.00; 3.32), 3.74 (0.62; 14.32) and 1.67 (1.04; 12.87) cell/ml. Number of integrin β4+ CTCs was significantly higher in luminal B (HER2+) and TNBC patients compared to luminal A/B (HER2−) ($p = 0.0084$ and $p = 0.0401$, respectively) (Fig. 3).

In-depth analysis of stem and EMT features by evaluation of the protein expression of CD133, ALDH1A1, CD44/CD24 and N-cadherin in integrin β4+ CTCs revealed the following patterns. There were minimal quantity CD44+CD24- stem CTCs expressing integrin β4+ in peripheral blood of breast cancer patients. So, significant differences in the number of integrin β4+ CTCs in the blood of HER2−positive were due to a higher content of cells with CD133 and ALDH1A1 stem markers ($p = 0.0072$ and ($p = 0.0094$), but not CD44+CD24- ($p = 0.0081$). Increase of integrin β4+ CTCs number in TNBC patients was provided by CD44+/CD24- ($p = 0.0388$) and CD133 ($p = 0.0442$) non-stem cell phenotypes. Concurrently, the number of CD133+ CTCs with integrin β4 expression in TNBC patients was significantly lower compared to luminal (HER2+) ($p = 0.0378$). The number of integrin β4+ CTCs during EMT (N-cadherin+) was higher in peripheral blood of patients with luminal B (HER2+) breast cancer ($p = 0.0087$).

Next, we evaluated whether there is a difference in the stem and EMT properties of integrin β4+ CTCs in metastatic breast cancer patients (Fig. 4).

It turned out that the number of integrin β4+ CTCs was higher in metastatic breast cancer patients ($p = 0.039$). The number of CTCs independently of expression of all studied markers was predominant among metastatic patients, but only non-stem by CD44+CD24-, without features of EMT (N-cadherin-) and during EMT (N-cadherin+) reached significant values ($p = 0.021$, $p = 0.029$ and $p = 0.033$, respectively).

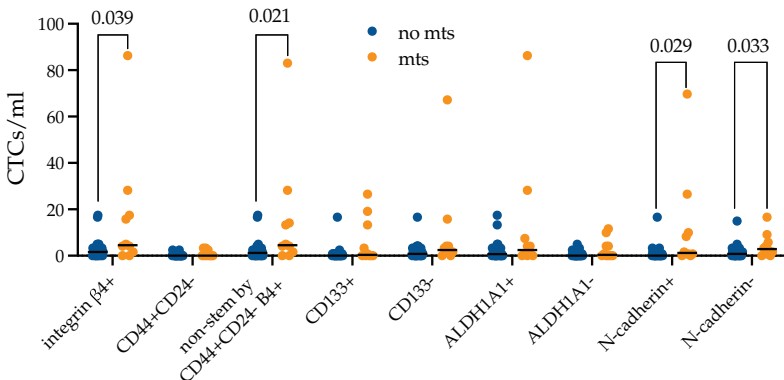

**Figure 4 Stem and EMT properties of integrin β4+ CTCs in metastatic breast cancer patients.** The number of integrin β4+ CTCs was higher in metastatic breast cancer patients ($p = 0.039$). The differences were unrelated to N-cadherin expression and were provided by non-stem by CD44+CD24- ($p = 0.021$).

## DISCUSSION

Signal transduction mediated by integrins upon binding to ECM ligands regulates a wide range of intracellular events covering both normal physiological and pathological processes (*Su et al., 2020*). There is evidence that in suspended epithelial cells, the function of integrins is suppressed due to the absence of ligand molecules (*Chen & Debnath, 2013*). However, this paradigm does not fully reflect the complex nature of the functioning of the integrin axis. Emerging evidence suggests that alternative mechanisms of integrin activation may be triggered by internal signals. Disruption of cell-ECM adhesion results in the dissociation of integrins from their ligands, consequently depriving cells of growth-promoting signals emanating from the ECM. Nonetheless, integrin activation can be induced through cytoskeletal rearrangements, where actin filaments recruit integrin adapter proteins such as talin, kindlin, paxillin, and p130, thereby orchestrating integrin clustering and subsequent activation. The activated integrin, in turn, initiates a cascade of events, including the activation of the FAK/SFK pathway and downstream signaling effectors. Consequently, the activated integrin signaling pathway exerts regulatory control over diverse forms of cellular demise, encompassing anoikis, autophagy, and entosis as well as cell cycle arrest thereby enabling anchorage-independent cell survival (*Deng et al., 2021*).

Since the functions of integrins are mainly related to heterodimeric form, it was expected that genes of complementary subunits that can provide synthesis of the corresponding heterodimers of subunits (in case of successful translation) should probably be expressed in CTCs simultaneously. The functional significance of the expression of non-complementary subunits is unclear since the description of subunit functions more often implies the heterodimers in which it is included. Adversely, the non-complementary integrin subunit expression is a physiological process, since according to the results of the database GSE168838 analysis the same variants of integrin subunits and heterodimer expression as in the studied CTCs were observed in luminal non-tumor cells of mammary

gland acini (*Raths et al., 2023*). There were only quantitative but not qualitative differences in integrin subunit expression between normal epithelial cells and CTCs.

In our study, 455 CTCs from scRNA-seq dataset were divided into nine groups depending on the combination of integrin gene expression. There was no integrin gene expression in 154 CTCs, whereas 224 CTCs (groups 1 and 2) expressed genes of non-complementary integrin subunits that could not form a full-fledged heterodimer. Two cells belonging to group 4 expressed only the genes of complementary subunits that could potentially form heterodimers. In 65 cells of groups 3, 5–8 non-complementary subunits were expressed in varying amounts along with subunits that can form heterodimers. Group 4 included two cells expressing only heterodimers.

As for the number of integrin heterodimers potentially expressed in each cell, the following pattern was observed. Five CTCs belonging to group 8 with *ITGA6* and *ITGB4* expression were characterized not only by the greatest number of potential heterodimeric receptors (from 3 to 7), but also by their greatest diversity: α6β4, αvβ1, αvβ5 и α2β1, α6β1, αVβ1, α5β1. In contrast, 60 CTCs belonging to groups 3–7 without *ITGA6* and *ITGB4* expressed integrin genes which potentially could form 1 or 2 heterodimeric receptors with less diversity of heterodimeric variants. At the same time, there were no CTCs with expression of integrins α4β1, αMβ2, αLβ2, α2bβ3 in group 8.

Evidently, anchorage-independent survival serves as a distinctive trait exhibited by all CTCs, facilitating the colonization of distant organs by tumor cells. While the role of integrins for communication in attached cells is relatively well understood, their role in the CTCs raises many questions. As mentioned above, in the epithelial cells that have lost cell communication, integrins can be activated by internal cellular signals.

In this regard, our initial hypothesis was that the functional activity of CTCs may be associated with integrins expressed in the cells. We used two approaches to identify the CTC functional potencies. The first one was the comparison of Top50 differentially expressed genes in the studied groups compared to the no integrins CTCs. Secondly, we evaluated the frequency of expression of 36 tumor progression-related genes in each group compared to the no integrins CTCs (Tables S2–S5). Since there is no linear correlation between the level of gene expression and the functionally significant amount of synthesized protein, the fact of positive gene expression is sufficient for the assumption of potency for a particular cell function.

It turned out that the most numerous CTCs with no integrin expression (group 9) had the smallest spectrum of functional activities. Thus, the positive expression of 26 out of the 36 evaluated genes occurred in CTCs with a frequency not exceeding 4%, while the expression of 19 genes was absent. Among the CTCs without integrin subunit expression, there was a low proportion of cells with features of stemness, no cells with laminin, chemokines/proinflammatory cytokines and metalloproteases expression (Tables S2–S5).

Evaluation of the differential gene expression revealed that only CTCs with *ITGA6* and *ITGB4* expression demonstrated increased expression of 14 tumor-progression related genes compared to the no integrins CTCs, almost half of which were related to EMT. According to the second criterion, we assessed the frequency of expression of 36 selected genes of interest in all groups of CTCs. The most significant differences (26 out of 36) were

also found in CTCs with *ITGA6* and *ITGB4* expression, while in CTCs belonging to groups 1–7, significant differences were found for 2 to 6 genes.

Moreover, the differences detected in CTCs with *ITGA6* and *ITGB4* compared to the no integrins CTCs, in which the greatest number and diversity of heterodimeric integrins have been detected, provide these cells with functional potencies to become seeds and cells contributing to the formation of metastatic niches. This is evidenced by the gene expression of laminin (only in these five cells out of 455), markers of stemness and hybrid EMT phenotype, markers of mesenchymal type of invasion, chemokine and proinflammatory cytokines.

We have validated the obtained data on integrin α6β4 gene expression on the protein level. There was no commercially available monoclonal antibody against heterodimeric form, so we used an antibody against integrin β4 subunit. Undoubtedly, this was a limitation of this part of the study; nevertheless, we were able to identify important associations of integrin β4 subunit expression with molecular type and hematogenous metastasis of breast cancer. The flow cytometry analysis revealed that the number of integrin β4+ CTCs was higher in patients with metastases, as well as more aggressive molecular subtypes of breast cancer (luminal (HER2+) and TNBC) compared to luminal A/B (HER2−) breast cancer. The discovered differences were due to the higher content of CD133+ and ALDH1A1+ CTCs, but not CD44+CD24-stem CTCs. These data indirectly confirm the results obtained by scRNA-seq. Indeed, we observed more frequent gene expression of *CD133* and *ALDH1A1*, but not *CD44+/CD24-* stem genes in polyfunctional CTCs with *ITGA6* and *ITGB4* gene expression (group 8) with presumed potency to develop distant metastases. Notably, there is evidence that integrin α6 may be the only biomarker commonly found in more than 30 different populations of stem cells, including some cancer stem cells, particularly in breast cancer (*Krebsbach & Villa-Diaz, 2017*).

What could explain such a high functional potential of CTCs with *ITGA6* and *ITGB4* expression? Integrin α6β4 is a heterodimeric receptor expressed on epithelial cells that binds specific laminin isoforms of the basal membrane and forms the core of hemidesmosomes (*Nishiuchi et al., 2006*). Tumor cells can adapt to the loss of adhesion to the basal membrane and avoid anoikis through α6β4-mediated adhesion to autocrine-produced laminin. As a result, such tumor cells acquire the ability to grow independently of their attachment to the basal membrane (anchorage-independent growth). It is believed that these changes contribute to tumor progression, including metastasis (*Bertotti, Comoglio & Trusolino, 2006*; *Zahir et al., 2003*; *Stewart & O'Connor, 2015*; *Beaulieu, 2019*). *Ramovs, Te Molder & Sonnenberg, 2016* summarized data on the function of laminin binding integrin α6β4 in the modulation of receptor tyrosine kinases (RTK) signaling pathways, including epidermal growth factor receptor family members EGFR and ErbB-2. This interaction stimulates prooncogenic signaling pathways, promoting nuclear translocation of transcription factors that ensure proliferation and survival of tumor cells (*Ramovs, Te Molder & Sonnenberg, 2016*). Moreover, such a surrogate basal membrane replacement provided by integrin α6β4-mediated laminin production supports anchorage-independent survival, not only of individual cells but also of cell complexes in 3D cultures (*Weaver et al., 2002*).

There is ample evidence that the most important functional states of cells (stemness, EMT, invasion, migration, proliferation, and tumor angiogenesis) are associated with α6β4 integrin expression (*Romagnoli et al., 2019*; *Bierie et al., 2017*; *Masugi et al., 2015*; *Yang et al., 2021*; *Roussellea & Scoazecb, 2020*). It is also known that α6β4 expression is associated with a poor prognosis in breast cancer patients (*Lu et al., 2008*). Thus, the available studies suggest that integrin α6β4 is indeed capable of providing anchor-independent resistance to apoptosis by binding to autocrinally synthesized laminin and activating signaling pathways associated with the metastatic potential of CTCs.

Initially, integrins were considered as very promising drug targets due to their essential functional role and extracellular localization of ligand-binding and regulatory sites. Despite disappointing results from clinical trials of anti-integrin therapies for solid cancers, these failures offer both opportunities and the need for improved therapeutic approaches.

A novel and promising approach to combat cancers involves combining integrin-targeted therapy with immunotherapy. For instance, integrin β4 has recently been proposed as an immunotarget in mouse models of mammary and head and neck tumors. Notably, the therapeutic efficacy of these ITGB4-targeted immunotherapies was significantly augmented when co-administered with anti-PD-L1, with no apparent systemic toxicity (*Ruan et al., 2020*). Hence, the identified subpopulation of tumor cells exhibiting simultaneous expression of ITGA6 and ITGB4 may serve as a viable target for ongoing therapeutic developments. Stratification of cancer patients based on prognostic biomarkers appears to be a promising avenue for optimizing existing drug therapies.

## CONCLUDING REMARKS

The analysis of available data and results obtained in our study allow us to draw the following conclusions and assumptions:

1) Among the studied CTCs, cells without *EPCAM* expression (97.4% of all CTCs) were predominant. CTCs in our study have extremely low proliferative activity according to *MKI67* expression (0.88%). Comparison of *SNAI1*, *SNAI2*, *ZEB1*, *ZEB2*, *CDH1*, *CDH2*, *EPCAM*, *KRT* and *VIM* gene expression in each group of CTCs suggest that almost all CTCs have different variants of hybrid EMT phenotypes.

2) CTCs were characterized by pronounced heterogeneity in the expression of integrin genes. CTCs without the expression of integrins or with the expression of non-complementary α and β subunits that cannot form heterodimers prevailed in the blood of breast cancer patients.

3) The number of different heterodimeric integrins in CTCs was associated with the gene expression of integrin α6β4: in the presence of α6β4 expression, there are three to seven different heterodimers in each CTC: α6β4, αvβ1, αvβ5, and α2β1, α6β1, αVβ1, α5β1; in the absence of integrin α6β4, there are only 1–2 heterodimers per cell, while the set of heterodimers was predominantly different – α4β1, αMβ2, αLβ2, α2bβ3.

4) Apparently, the anchor-independent survival is inherent to CTCs which could become potential metastatic seeds and form distant metastases. The resistance to apoptosis in such CTCs could be achieved in different ways: a) by switching the external activation of

integrins when they bind to ECM ligands to the mechanism of internal activation, b) by synthesis of basal membrane proteins by the epithelial cell which have lost their adhesion with the basal membrane.

5) Probably, the second variant of anchor-independent survival is typical for the CTCs potentially expressing the heterodimeric integrin α6β4 and several laminin subunits (other CTCs do not express genes of laminins).

6) According to obtained data CTCs expressing genes of integrin α6β4 have the highest functional potencies. These cells were characterized by the largest number and diversity of heterodimeric integrin gene expression, more frequent expression of stem-related genes, genes associated with the mesenchymal type of invasion, genes of chemokines and pro-inflammatory cytokines compared to the no integrins CTCs. The combination of these functions indicates a significant potential of integrin α6β4-expressing CTCs to become the disseminated cells and act as metastatic seeds and/or cells that promote the formation of metastatic niches.

7) In comparison with CTCs expressing integrin α6β4 genes, CTCs expressing any non-complementary integrin subunits or other heterodimeric integrins do not differ from the no integrins CTCs by expression of the studied functional genes.

8) The number of integrin β4+ CTCs expressing CD133 and ALDH1A1 stem markers but not CD44+CD24- was increased in metastatic breast cancer as well as in molecular subtypes of breast cancer with poor prognosis (luminal (HER2+) and TNBC) compared with luminal A/B (HER2−) subtype.

9) The hypothesis describing the functional status of integrin α6β4-expressing CTCs can be represented as follows: the integrin α6β4 initiates autocrine synthesis of laminins with following binding, reproducing an activation similar to the interaction with the basal membrane. Because of this, the cell implements a stable mechanism of anchor-independent survival. Other full-length heterodimeric integrins can be expressed in parallel, resulting in the activation of many signaling pathways. Finally, various signaling pathways are activated that provide this polyfunctional state of α6β4-expressing CTCs.

The interpretation of the results has a number of limitations. The assumption that expression of the genes of complementary integrin subunits will be realized in the translation and assembly of complete heterodimeric integrins is not indisputable, but only probable.

The conclusions concern mainly the luminal molecular subtypes of breast cancer, which have the lowest rate of distant metastasis. In fact, 3 years follow-up established the development of distant metastases in only one of the 20 patients. Therefore, it was not possible to identify the association of data obtained by scRNA-seq with distant metastasis.

## CONCLUSIONS

Single-cell sequencing of CTCs revealed heterogeneity of integrin gene expression. The most numerous groups of CTCs either lacked expression of any integrins or showed expression of various non-complementary integrin subunits that theoretically cannot form

a complete heterodimeric receptor. Among CTCs with the gene expression of complementary integrins, cells with one or two heterodimers were predominant. CTCs expressing the integrin α6β4 genes were identified in which three to seven other different heterodimeric receptors were simultaneously expressed. In order to elucidate the association of different variants of integrin expression in CTCs with their potential functional activity, both differential gene expression and the frequency of 36 tumor progression-related gene expression that are relevant to stemness, EMT, invasive potency, and the ability to secrete cytokines were determined. The lowest frequency of gene expression was observed in CTCs without integrin expression. The same pattern was detected for CTCs with expression of non-complementary integrin subunits or CTCs without co-expression of *ITGB4* and *ITGA6*, which was characterized by altered expression of several genes. Only the *ITGB4* and *ITGA6*-expressed CTCs showed differentially expressed genes and the maximum number of increased tumor progression-related genes (26/36) compared to the no integrin CTCs. The presence of differentially expressed genes and the high frequency of gene expression of laminins, EMT- and stemness-related genes, markers of invasion and proinflammatory cytokines/chemokines in α6β4-expressing CTCs suggests their polypotent functional activity associated with anchorage-independent survival, which is ensured by the interaction of α6β4 integrin with autocrine synthesized laminins. It is reasonable to assume that it is these CTCs that correspond most closely to the properties of metastatic seeds and/or cells involved in the formation of pre-metastatic niches. The result of assessing the number of CTCs with integrin β4 protein expression confirms the association of this integrin with metastasis in breast cancer. Studies on large groups of patients with different molecular subtypes of breast cancer, could overcome the limitations of our study and will clarify the association of integrin expression in CTCs with their functional activity. It seems particularly important to clarify the role of anchorage-independent survival mediated by interaction of integrin α6β4 with autocrine laminins providing a high degree of polyfunctionality and ability to initiate the development of hematogenous metastases.

### Funding
The study was supported by the Russian Science Foundation (grant #21-15-00140). The funders had no role in study design, data collection and analysis, decision to publish, or preparation of the manuscript.

### Grant Disclosures
The following grant information was disclosed by the authors:
Russian Science Foundation: #21-15-00140.

### Competing Interests
The authors declare that they have no competing interests.

## Author Contributions

- Evgeniya Grigoryeva conceived and designed the experiments, obtained and analyzed the flow cytometry data, interpretation of scRNA-seq data, prepared figures and/or tables, and approved the final draft.
- Liubov Tashireva conceived and designed the experiments, interpretation of scRNA-seq data, prepared figures and/or tables, and approved the final draft.
- Vladimir Alifanov performed the flow cytometry experiments, prepared figures and/or tables, and approved the final draft.
- Olga Savelieva performed the flow cytometry experiments, prepared figures and/or tables, and approved the final draft.
- Marina Zavyalova performed the experiments, authored or reviewed drafts of the article, and approved the final draft.
- Maxim Menyailo performed sample preparation, scRNA-seq procedure performing, bioinformatic analysis, prepared figures and/or tables, and approved the final draft.
- Anna Khozyainova performed sample preparation, scRNA-seq procedure performing, bioinformatic analysis, prepared figures and/or tables, and approved the final draft.
- Evgeny V. Denisov performed bioinformatic analysis, authored or reviewed drafts of the article and approved the final draft.
- Olga Bragina performed the experiments, prepared figures and/or tables, treatment and inclusion of patients in the study, and approved the final draft.
- Nataliya Popova analyzed the data, prepared figures and/or tables, treatment and inclusion of patients in the study, and approved the final draft.
- Nadezhda V. Cherdyntseva conceived and designed the experiments, authored or reviewed drafts of the article, and approved the final draft.
- Vladimir Perelmuter conceived and designed the experiments, analyzed the data, authored or reviewed drafts of the article, and approved the final draft.

## Human Ethics

The following information was supplied relating to ethical approvals (*i.e.*, approving body and any reference numbers):

The study was approved by the local ethics committee of the Tomsk NRMC Institutional Review Board (17 June 2016, the approval No. 8), and informed consent was obtained from all patients prior to analysis.

## Data Availability

The sequence reads are available to NCBI: PRJNA776403.

## Supplemental Information

Supplemental information for this article can be found online at http://dx.doi.org/10.7717/peerj.16678#supplemental-information.

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
