# Peer review of "Integrin-associated transcriptional characteristics of circulating tumor cells in breast cancer patients"

_PeerJ, doi:10.7717/peerj.16678_

## Round 0.1 · original submission · Minor Revisions

The article holds merit and interest to the journal's readership due to the findings that the co-expression of ITGA6 and ITGA4 in CTCs is associated with EMT and stemness-related genes. The authors are requested to address the concerns of the reviewers, especially regarding the statistical analysis and sample size. Additionally, please have a fluent English speaker review the manuscript to improve readability before resubmission.

**Language Note:** The Academic Editor has identified that the English language must be improved. PeerJ can provide language editing services - please contact us at copyediting@peerj.com for pricing (be sure to provide your manuscript number and title). Alternatively, you should make your own arrangements to improve the language quality and provide details in your response letter. – PeerJ Staff

Reviewer 1 ·

Basic reporting

Overall, it is nice that the authors found that CTCs co-expressed ITGA6 and ITGA4 are more related to the EMT and stemless-related genes. However, the writing is not ideal and the paper is not easy to read. The authors have to work on the English to get the article published.

Experimental design

1. In the second part of the results, the authors showed that the over expressed genes were involved in the EMT process. Are there any other signaling pathways detected? Could the author show the result by gene enrichment analysis or GSEA? It would always be better to visualize the results rather than explaining it by plain language.
2. Moreover, the authors showed the group that express ITGA6 and ITGB4 is quite different compared to the other groups in terms of gene expression. Can you plot the genes in the figure to show the difference?
4. Figure 1 and figure 2, how many samples are analysed? Could you show each samples in the plot rather than the median value for each bar? What is the statistic method used here?
4. Please check the other conclusion you made as well. Please include a figure if possible.

Validity of the findings

No comments

Reviewer 2 ·

Basic reporting

1. Authors are advised to consolidate all supplementary tables into one PDF file for readers' convenience.

2. Certain acronyms, such as EMT and NAC, aren't defined within the main text. To enhance clarity, especially for readers unfamiliar with the subject, please introduce the full term before using its acronym.

3. Regarding Figures 1 and 2, it's crucial to elucidate the method used to compute the p-value. If p-values are absent, it's presumably because they exceeded 0.05; please confirm this in the figure legend. There are also instances of missing data, notably with CD44+CD24-'s CTCs in Figure 1, and 'no mts' data in several conditions. Both the main text and figure legends should account for this missing information.

Experimental design

The "Materials & Methods" section is lucidly presented, with a solid experimental framework. A minor enhancement would be to specify the release version of the RNASeqPower tool when discussing statistics. This detail would bolster the reproducibility of your methods.

Validity of the findings

1. In the concluding remarks section, nine conclusions and assumptions are outlined. To amplify their relevance, it would be beneficial to illustrate how these conclusions reinforce the paper's core proposition: the potential correlation between the functional potency of CTCs and the risk of distant metastases. This approach would make the paper's insights more accessible and impactful for readers.

2. Furthermore, positioning this study within the broader academic landscape is advisable. Discussing any analogous research and comparing this study's findings with others will accentuate its uniqueness and importance. After highlighting the relationship between integrin ⍺6β4 and autocrine laminins in connection to hematogenous metastases, it would be enriching to suggest potential avenues for future research, especially regarding therapeutic interventions targeting this newly discovered pathway.

---

## Round 0.2 · Minor Revisions

The revised manuscript has addressed all the reviewer comments and the authors have improved the revision significantly. One minor correction to improving the readability as provided by the reviewer is to add more detailed figure legends instead of putting in the context. The authors are requested to make this revision.

Reviewer 1 ·

Basic reporting

The overall writing has been greatly approved with the new submission and the authors had answered all my concerns. the only comment to add is that it would be clear if the authors can add more detailed figure legends instead of putting in the context.

Experimental design

no comments.

Validity of the findings

no comments.

Reviewer 2 ·

Basic reporting

The authors have thoroughly addressed the feedback I provided on basic reporting.

Experimental design

The authors incorporated my suggestions by adding a detailed basis for their RNA sequencing analysis.

Validity of the findings

The authors adopted my suggestions, thereby amplifying the manuscript's impact by offering a prospective discussion on the potential of their research in synergizing integrin-targeted therapy with immunotherapy.

---

## Round 0.3 · accepted · Accept

The remaining concerns were adequately addressed and the revised manuscript is acceptable now.